# Preparation of Amino-Functional UiO-66/PIMs Mixed Matrix Membranes with [bmim][Tf_2_N] as Regulator for Enhanced Gas Separation

**DOI:** 10.3390/membranes11010035

**Published:** 2021-01-04

**Authors:** Jiangfeng Lu, Xu Zhang, Lusheng Xu, Guoliang Zhang, Jiuhan Zheng, Zhaowei Tong, Chong Shen, Qin Meng

**Affiliations:** 1Center for Membrane and Water Science &Technology, Institute of Oceanic and Environmental Chemical Engineering, State Key Lab Breeding Base of Green Chemical Synthesis Technology, Zhejiang University of Technology, Hangzhou 310014, China; jln106412@sina.com (J.L.); xz669592@sina.com (X.Z.); xulusen@zjut.edu.cn (L.X.); zhengjiuhan@163.com (J.Z.); lzw199168@163.com (Z.T.); 2Department of Chemical and Biological Engineering, State Key Laboratory of Chemical Engineering, Zhejiang University, Hangzhou 310027, China; rainbows@zju.edu.cn

**Keywords:** mixed matrix membrane, amino-functional UiO-66, PIM-1, [bmim][Tf_2_N], gas separation

## Abstract

Development of mixed matrix membranes (MMMs) with excellent permeance and selectivity applied for gas separation has been the focus of world attention. However, preparation of high-quality MMMs still remains a big challenge due to the lack of enough interfacial interaction. Herein, ionic liquid (IL)-modified UiO-66-NH_2_ filler was first incorporated into microporous organic polymer material (PIM-1) to prepare dense and defect-free mixed matrix membranes via a coating modification and priming technique. IL [bmim][Tf_2_N] not only improves the hydrophobicity of UiO-66-NH_2_ and facilitates better dispersion of UiO-66-NH_2_ nanoparticles into PIM-1 matrix, but also promotes the affinity between MOFs and polymer, sharply reducing interface non-selective defects of MMMs. By using this strategy, we can not only facilely synthesize high-quality MMMs ignoring non-selective interfacial voids, but also structurally regulate MOF nanoparticles in the polymer substrate and greatly improve interface compatibility and stability of MMMs. The method also gives suitable level of generality for fabrication of versatile defect-free MMMs based on different combination of MOFs and PIMs. The prepared UiO-66-NH_2_@IL/PIM-1 membrane exhibited outstanding gas separation behavior with large CO_2_ permeation of 8283.4 Barrer and high CO_2_/N_2_ selectivity of 22.5.

## 1. Introduction

With the development of the economy, the soaring increase of greenhouse-gas emissions generated from industrial activities poses a huge threat to human society. Massive emissions of greenhouse-gases like CO_2_ have profoundly affected the lives and future of people. CO_2_ capture and storage are receiving widespread attention around the world for environmental reasons [1,2]. The currently used CO_2_ removal methods, such as cryogenic liquefaction and solvent absorption, usually require large amounts of energy consumption and cause environmental pollution. Compared with these conventional processes, membrane separation technology has huge advantages due to its simple process operation and efficient separation. At present, polymer-based membranes are involved in most applications of membranes in gas separation on account of their good processability, low price and easy availability [2]. Unfortunately, the use of polymer membranes is subject to the trade-off between selectivity and permeability, known as the Robeson upper bound [3,4]. Exploration of new polymer materials with high efficiency for CO_2_ removal is very critical for industrial separation.

As a new kind of microporous organic materials with excellent thermochemical stability and high free volume polymers of intrinsic microporosity (PIMs) have attractive prospects in gas separation [5]. PIMs consist of light elements C, N, H and O, which have better solubility and processability than conventional microporous polymers. PIM-1, one of the most representative PIMs materials, can be used in the preparation of continuous membranes for gas separation [6,7]. To improve the characteristics of pristine PIM-1, a range of derivatives were designed and synthesized on the basis of PIM-1 by replacing the rigid units of the main chain or post-modification of the side groups to improve the separation effect [5,8]. However, the use of pure PIM-1 membrane is still very restricted by some inherent defects of the polymers such as irregular pore size distribution and unstable dynamic properties [9], leading to a significant separation efficiency insufficiency.

To increase the practicability of polymer membranes based on polymers of intrinsic microporosity, the design concept of introducing porous fillers into the polymer matrix for preparing high-quality mixed matrix membranes (MMMs) is proposed [10,11]. This construction can not only maintain the good processability of the polymer, but also combine the superior transport capacity of the fillers [12,13,14]. So far, 2D materials such as C_3_N_4_ nanosheets, graphene oxide, nanoparticle SiO_2_ and other porous fillers have been successfully introduced into PIM-1 materials, which have improved the permeability or selectivity of membranes to some extent [15,16,17]. Compared with the above materials, the emerging metal organic frameworks (MOFs) may be one of the best filler options. According to our experience, MOFs are porous crystalline materials that are easy to synthesize and have diverse structures [18,19]. Further, their adjustable pore size enables MOFs to effectively distinguish between different gas molecules, and the large specific surface area allows MOFs to have more adsorption sites than other materials [9,20]. As reported, a MMM was successfully prepared by mixing ZIF-71 with PIM-1 for CO_2_/CH_4_ separation. By incorporation of ZIF-71 and UV treatment, the synthesized membrane achieved a high CO_2_ permeability up to 3458.6 Barrer with relatively high gas selectivity [21]. Similarly, a high performance MMM was prepared based on ZIF-67 and PIM-1 with CO_2_ permeation value of 4521 Barrer and CO_2_/CH_4_ selectivity of 24.2, both far exceeding the effects of pure PIM-1 [9]. Although these progresses show that the combination of appropriate MOF and PIM-1 matrix may be a feasible strategy for developing high-performance gas separation MMM, some issues still need to be addressed during the MMM design and preparation process. For example, the poor adhesion between filler and matrix tends to form defects and non-selective interfacial voids, which may severely degrade membrane separation efficiency [22,23]. Polymer hydrophobicity and filler hydrophilicity tend to exacerbate this phenomenon [24]. Meanwhile, the uneven dispersion of the fillers will cause a serious aggregation tendency to reduce the surface energy, which may severely limit the contents of the filler in MMM or even destroy the membranes completely [25].

To increase the affinity of MOF and polymers and reduce non-selective defects, different methods have been tried to improve the interface interaction between the two phases [23,26,27]. Since nano-sized filler particles can provide a larger interfacial contact area and exhibit stronger interactions with the polymer matrix, applying a smaller MOF or limiting the size of the MOF filler may be an effective way. For example, after reducing the size of UiO-66 by functionalizing MOF walls and water modulation, MOF particles of 20–30 nm were synthesized and a prepared MMM with better separation efficiency was obtained [10]. Another way to improve the adhesion of polymers to MOF is to modify the outer surface of a MOF with functional groups that can form strong or weak interactions with specific polymers. By applying amine-functionalized MIL-53 as the MMM dispersed phase, the amine groups of MIL-53 interacted with the polymer through hydrogen bonds, which adjusted the spatial distribution of MOF particles and reduced non-selective interfacial voids [22]. Recently, the incorporation of ionic liquids (ILs) into the filler/polymer interface has been explored. Ionic liquids have attractive physical and chemical properties, such as high thermochemical stability and extremely low volatility [28,29]. As good gas separation material, ionic liquids not only have high CO_2_ solubility, but also can be directly introduced into the membrane as a two-phase lubricant for polymers and fillers [30,31]. By introducing [BMIM][BF_4_] on the basis of the original binary Pebax 1657/Ag, a new ternary MMM was prepared. With the newly introduced IL, Pebax 1657 matrix and Ag nanoparticles displayed tighter interfacial adhesion [32]. Therefore, the introduction of ionic liquids is an effective solution to break through the interfacial bottlenecks between polymers and fillers. 

In this work, we report a new and facile method of coating modification and priming technique to synthesize high-quality amino-functional UiO-66/PIM-1 mixed matrix membrane with ionic liquid [bmim][Tf_2_N] as regulator for enhanced CO_2_ separation. UiO-66-NH_2_ particles were modified by solvent immersion and the surface of UiO-66-NH_2_ was coated with [bmim][Tf_2_N]. Then the UiO-66-NH_2_@IL was mixed with PIM-1 matrix by priming technique to prepare dense and defect-free MMMs (Figure 1). Our strategy has the following advantages: (1) The selected IL [bmim][Tf_2_N] can act as a regulator of PIM-1/UiO-66-NH_2_, thereby promoting the uniform distribution of filler particles in MMMs. (2) Amino-functional UiO-66 with abundant micropore structure can offer many additional pathway for CO_2_ permeation and form hydrogen bonds with PIM-1 to improve interfacial interactions and reduce the non-selective defects of MMMs. (3) Both introduced IL and amino group have strong affinity for CO_2_, can be used as a selective CO_2_ transport carrier for better separation of MMMs. To our knowledge, so far there is no report on the preparation of high-quality PIM-based MMMs using IL as a regulator to eliminate the interface defects. Moreover, based on the forming mechanism of our strategy, different metal organic frameworks and polymers of intrinsic microporosity can be employed using ionic liquid as regulators to prepare versatile dense and defect-free MMMs. Further, the prepared mixed matrix membranes displayed excellent gas separation behavior.

## 2. Materials and Methods

### 2.1. Materials

1-Butyl-3-methylimidazolum bis(trifluoromethylsulfonyl)imide ([bmim][Tf_2_N]) and zirconium (IV) chloride (ZrCl_4_) were obtained from Aladdin Chemical Reagent Co. (Shanghai, China). 5,5′,6,6′-Tetrahydroxy-3,3,3′,3′-tetramethyl-1,1′-spirobisindane (TTSBI, 97%) was purchased from Alfa Aesar (Hangzhou, China). 2,3,5,6- Tetrafluoroterephthalonitrile (TFTPN, 99%) was purchased from Sigma Aldrich (Hangzhou, China). 2-Aminoterephthalic acid (H_2_bdc-NH_2_, 99%) was purchased from TCI (Hangzhou, China). Methanol, dimethylacetamide (DMAc), chloroform (CHCl_3_), N,N-dimethylformamide (DMF), toluene, potassium carbonate (K_2_CO_3_) and acetic acid were obtained from Sinopharm Chemical Co. Ltd. (Hangzhou, China).

### 2.2. Synthesis of PIM-1

PIM-1 was synthesized by a high-temperature method, similar to previous report [33]. In terms of details, TFTPN (3.00 g), TTSBI (5.10 g) and anhydrous K_2_CO_3_ (6.20 g) were added to DMAc (40 mL) in the flask and stirred, 20 mL toluene was subsequently added. The polymerization process was maintained at 160 °C. After the reaction, the viscous liquid was poured into methanol to obtain a yellow thread-like polymer which was then further purified by chloroform and methanol. The collected product was washed with deionized water, after which the prepared product was dried under vacuum before use.

### 2.3. Synthesis of UiO-66-NH_2_

The synthesis of MOF was carried out as previously reported [34]. H_2_bdc-NH_2_ (376 mg) and ZrCl_4_ (488 mg) were added into DMF (120 mL) and stirred. 3.6 mL acetic acid was dropped subsequently. The reaction mixture was kept at 120 °C. After the reaction, the light-yellow precipitate was collected and washed. Then the final product was dried for further use.

### 2.4. Synthesis of UiO-66-NH_2_@IL

A facile solvent immersion method was used to prepare UiO-66-NH_2_@IL [35]. [bmim][Tf_2_N] (2.5 g) was added to methanol (50 mL) and stirred. Then 0.5 g UiO-66-NH_2_ particles were added and stirred at 50 °C. The mixture was then centrifuged. Finally, the collected UiO-66-NH_2_@IL particles were dried at 80 °C overnight.

### 2.5. Membrane Preparation

PIM-1 was first dissolved in a vial of chloroform to obtain a fluorescent yellow solution. Then a certain amount of UiO-66-NH_2_ (or UiO-66-NH_2_@IL) was dispersed in another vial of chloroform to obtain a MOF slurry. PIM-1 and MOF were mixed by means of priming technique [36]. Specifically, the filler was primed by adding a small amount of PIM-1 solution. After stirring, the remaining PIM-1 solution was then added to the mixture. The PIM-1/MOF mixed solution was further stirred overnight. The mass fraction of the filler in the mixed solution was defined by Equation (1):(1)$$Filler\;loading\;(wt\%)=\frac{MOF\;wt}{MOF\;wt\;+\;polymer\;wt}\times 100\%$$

Before casting, the finally obtained MOF/PIM-1 mixture was sonicated to avoid air bubbles and then poured into a Teflon mould to form a homogeneous membrane. The filler loading in MMMs was 5, 10, 20, 30 wt.%, respectively. All prepared membranes were kept at room temperature to evaporate the solvent. Subsequently, the obtained MMMs were dried under vacuum at 80 °C. For easy identification the membrane made of UiO-66-NH_2_@IL was designated as MOF@IL/PIM-1 and the membrane made of unmodified UiO-66-NH_2_ was designated as MOF/PIM-1. Controlled experiments were also conducted including pure PIM-1 membrane and IL/PIM-1 MMM (IL as filler).

### 2.6. Characterization Techniques

Morphology of samples and cross-section images of MMMs were observed by field emission scanning electron microscopy (SEM, HIYACHI SU8010, Tokyo, Japan). The composition and distribution of elements in the samples were analyzed by energy dispersive X-ray spectrometer (EDX, HIYACHI SU8010). All samples required to be sprayed with gold in vacuum before testing. Elemental content and difference of UiO-66-NH_2_ and modified UiO-66-NH_2_@IL were measured by X-ray photoelectron spectroscopy (XPS, PHI 5300, Shanghai, China). The crystalline structure of MOFs was measured by X-ray diffraction (XRD, X’Pert Pro, Hangzhou, China). The introduction of IL and the membrane structure were measured by Fourier transform infrared spectroscopy (FT-IR, Nicolet 6700, Thermo Scientific, Shanghai, China). Nitrogen adsorption-desorption of the UiO-66-NH_2_ and IL-modified UiO-66-NH_2_ were measured using a Quantachrome instrument (Hangzhou, China). A contact angle meter (OCEA15EC, Dataphysics, Shanghai, China) was used to characterize the hydrophobicity of the MOF sample. Gel permeation chromatography test (GPC, Waters Corporation, Shanghai, China) was used to characterize PIM-1. Polystyrene was used as the standard sample and tetrahydrofuran (1 mL min^−1^) was used as the mobile phase. The temperature of column and detector were 30 and 35 °C, respectively. Nuclear magnetic resonance (NMR, Bruker, Zurich, Switzerland) spectroscopy was used to characterize the structure of the polymer and the samples were dissolved in deuterated chloroform (CDCl_3_). Thermal gravimetric analysis (TGA) was performed at a rate of 10 °C/min under a nitrogen atmosphere using a TGA 7 instrument (Perkin Elmer, Shanghai, China).

### 2.7. Gas Permeation Tests

Gas permeability of the membrane was measured on a membrane performance evaluation unit by a standard variable volume method [37,38]. The membrane was placed in a performance evaluation unit and set at 20 °C with a pressure of 1 bar. Each membrane was tested at least three times to ensure the accuracy of the data. The permeability coefficient (*P*) is calculated using the following equation:(2)$$p=\frac{QL}{A(p_{u}-p_{d})}$$
where *Q* is the volume flow rate of gas (cm^3^[STP]s^−1^), *p_u_* and *p_d_* are the upstream and downstream pressures (cmHg) through the membrane, *L* refers to the membrane thickness (cm) and *A* refers to effective area (cm^2^), selectivity (α) is the ratio of the two gas permeabilities:(3)$$\alpha =\frac{p_{1}}{p_{2}}$$

## 3. Results and Discussion

### 3.1. Incorporation of IL in UiO-66-NH_2_

UiO-66-NH_2_@IL was obtained by blending UiO-66-NH_2_ and [bmim][Tf_2_N] via solvent-immersion method. As shown in Appendix A, the pristine UiO-66-NH_2_ exhibited agglomerated octahedral crystals with an average size of 100–200 nm. By comparing with related literature, the obtained UiO-66-NH_2_ is consistent with the reported UiO-66 characteristic peak position, and the crystal structure has not changed significantly [20]. After the introduction of IL, the polyhedral structure of crystal was unchanged (Figure 2a). Moreover, according to XRD patterns (Figure 2b), the crystalline lattice structure of UiO-66-NH_2_@IL agreed with the unmodified MOFs. The decrease in the relative intensity of the (111) peak indicated a significant change in electron density in the pores of the MOF crystal, demonstrating the introduction of the [bmim][Tf_2_N]. This was also consistent with the results reported in the relevant literature [28,35]. FT-IR was used to further confirm that IL was successfully incorporated on the MOF surface. Compared to unmodified UiO-66-NH_2_, IL modified samples showed some new characteristic peaks. As shown in Figure 2c, new absorption sites appear at 1063 cm^−1^, 1203 cm^−1^ and 1354 cm^−1^ corresponding to SNS asymmetric vibration, SO_2_ symmetric vibration and SO_2_ asymmetric vibration in the [Tf_2_N](-) anion, respectively. In addition, the spectrum at the high wavenumber (2800–3200 cm^−1^) indicated the presence of [bmim](+) cation [3,28]. The N_2_ absorption/desorption results both showed a type I isotherm (Figure 2d), which proved that the introduction of ionic liquid did not greatly change the micropore characteristics of MOFs. As shown in Table 1, compared with the original sample, the BET area of UiO-66-NH_2_@IL decreased. Simultaneously, the pore size and pore volume of MOF were also reduced to different degrees. Since the ionic liquid covered the surface of the MOFs and occupied part of the pores, the surface area and pore channel of UiO-66-NH_2_ were regulated. It is generally believed that the additives with a given channel geometry will introduce new gas transmission channels for the originally dense PIMs membrane material. Then, the small-sized gas molecules can be easier to pass through the membrane, resulting in the increase of gas permeance. In addition, the pore size and volume decrease of UiO-66-NH_2_@IL maybe leading to increase the selectivity of UiO-66-NH_2_@IL/PIM-1 MMM.

To investigate the IL content of the MOF, XPS survey spectra were used to analyze the surface of the samples (Appendix A). After IL modification, the content of C, O and Zr elements was decreased. Accordingly, the N element content rose from 4.83 wt.% to 6.06 wt.%. The new F element accounted for 2.86 wt.% and the S element accounted for 1.75 wt.%. Through further calculation and analysis, the surface coverage of IL accounted for 10.52–11.47 wt.%. The dispersibility of IL on MOF surface was investigated by EDX elemental analysis (Appendix A). It was found that elements such as N, F, and S were uniformly distributed on the outer surface of MOF, indicating the uniform distribution of IL.

### 3.2. Characterization of PIM-1

FT-IR, 1NMR and GPC tests were used to verify the successful synthesis of polymer. It is commonly known that -CN is one of the characteristic functional groups of PIM-1. As shown in Appendix A, the infrared radiation absorbed at a wavenumber of 2240 cm^−1^ corresponded to the -CN stretching vibration [38]. The ^1^H-NMR results are shown in Appendix A, and structure of the synthetic material also agreed with that reported in our previous work [7]. As shown in Appendix A, the molecular weight (Mw) of the material was 212.3 kDa and the number-average molecular weight (Mn) was 73.4 kDa, which were much higher than those reported in other literature [7]. The polydispersity Mw/Mn was 2.89 (Appendix A).

### 3.3. Characterization of Mixed Matrix Membranes

Figure 3 shows the cross-sectional morphology of pure PIM-1 membrane, UiO-66-NH_2_/PIM-1 MMM (20 wt.%) and UiO-66-NH_2_@IL/PIM-1 MMM (20 wt.%). Pure PIM-1 membrane had very regular texture. After mixing with UiO-66-NH_2_, some obvious voids could be observed in UiO-66-NH_2_/PIM-1 MMM. Meanwhile, phase separation occurred in the membrane. MOF particles began to agglomerate and the filler and polymer were not tightly bound. In contrast, no obvious defects and agglomeration of fillers were observed in UiO-66-NH_2_@IL/PIM-1 MMM. UiO-66-NH_2_@IL was well distributed in PIM-1 matrix and was tightly combined with the polymer to form a defect-free and homogeneous whole. It indicated that the interface defects in UiO-66-NH_2_@IL/PIM-1 MMMs were effectively reduced by modification of IL. According to previous report, the difference in hydrophilicity and hydrophobicity is one of the reasons for the insufficient compatibility between the filler and polymer [24]. The combination of the hydrophobic polymer matrix and hydrophilic filler often leads to the generation of interface defects. The selected [bmim][Tf_2_N] contains hydrophobic ions, which could bind to hydrophobic PIM-1 better than hydrophilic fillers. Moreover, ILs are viscous, so they can act as a binder between the polymer and filler, enhancing the interface affinity. Based on the above results, we speculate that the IL coated on the surface of UiO-66-NH_2_ hydrophobically modified the filler and regulate the interfacial compatibility between PIM-1 and UiO-66-NH_2_, meanwhile, the uniformity of the dispersion of UiO-66-NH_2_ in MMMs was also improved.

To further study the influence of filler content on the morphology of membrane, Figure 4 shows cross-sectional images of UiO-66-NH_2_@IL/PIM-1 MMMs with different UiO-66-NH_2_@IL loadings. As the UiO-66-NH_2_@IL content increased, phase separation did not appear in UiO-66-NH_2_@IL/PIM-1 MMM. UiO-66-NH_2_@IL and PIM-1 were blended closely without interfacial voids and non-selective defects. Even with the content as high as 30 wt.%, a good distribution of UiO-66-NH_2_@IL could be observed in the PIM-1 matrix. This may be the result of interfacial regulating effect of IL. IL acted as a regulator and binder between the UiO-66-NH_2_ filler and the PIM-1 polymer, increasing the affinity between the two phases and promoting UiO-66-NH_2_ evenly distributed.

The chemical structure change of the membranes was first verified by infrared spectroscopy (Figure 5a). For pure PIM-1 membrane, the peak at 2240 cm^−1^ is assigned to the nitrile group [25]. The peak around 1300–1100 cm^−1^ can correspond to the ether bond in PIM-1 [9]. As the amount of UiO-66-NH_2_ increased, the tensile strength of the ether bond was significantly weakened. This could be due to the existence of hydrogen bonds between PIM-1 and the amine group of UiO-66-NH_2_@IL. In spectrum of UiO-66-NH_2_@IL/PIM-1 MMMs, the appearance of broad N-H band at 3200–3600 cm^−1^ as well as the C=O band at 1650–1710 cm^−1^ indicate the presence of MOF particles in PIM-1. After the amount of MOF increased, the peak of the amino groups on the MMM became more blunt and broad, which should be related to the hydrogen bond interaction [10]. The thermal stability of UiO-66-NH_2_, pure PIM-1 membrane and UiO-66-NH_2_@IL/PIM-1 MMM were tested under a nitrogen atmosphere (Figure 5b). TGA test shows that the pure PIM-1 membrane exhibits excellent thermal stability at a high temperature up to 450 °C, and it gradually decomposes after 450 °C, which may be related to the strong dipole interaction of the nitrile group in PIM-1 polymer chain. In contrast, compared with PIM-1, the thermal stability of UiO-66-NH_2_ particles is less insufficient. When heated to 800 °C, the mass loss of UiO-66-NH_2_ was close to 60% [39]. For mixed matrix membranes, UiO-66-NH_2_/PIM-1 MMM had similar thermal stability as pure PIM-1 membrane. After the introduction of IL, the thermal performance of UiO-66-NH_2_@IL/PIM-1 MMM was not greatly affected. Below 450 °C, the mass loss of UiO-66-NH_2_@IL/PIM-1 MMM remained below 10%. It proved that PIM-based membranes with incorporation UiO-66-NH_2_@IL can still maintain the good thermal stability.

### 3.4. Gas Separation Properties

To investigate the gas separation behavior for membrane, a series of gas separation measurements were performed. From the results of Figure 6a,b, all the PIM-1 based membranes have a high permeability for CO_2_. By continuously adding UiO-66-NH_2_@IL, the CO_2_ permeability has been further improved. In particular, the UiO-66-NH_2_@IL/PIM-1 MMM with 30 wt.% UiO-66-NH_2_@IL has a high CO_2_ permeability up to 10040.1 Barrer, which is a 42.2% increment over that of pure PIM-1 membrane (7061.9 Barrer). Furthermore, the selectivity of the membrane has also been greatly enhanced. When the loading is only 10 wt.%, the CO_2_/N_2_ selectivity of UiO-66-NH_2_@IL/PIM-1 MMM changes from 10.3 to 22.5 and the selectivity of CO_2_/CH_4_ changes from 6.3 to 12.3, compared to pure PIM-1 membrane. These phenomena might be due to the fact that IL provides stable MOF suspension solution and thus avoid pre-agglomeration filler-polymer interfacial defects [40]. As is known, PIM-1 material has a strong affinity for CO_2_, but the gas selectivity is insufficient. The intrinsic CO_2_-targeted affinity properties of UiO-66-NH_2_@IL can provide a facilitated CO_2_ transport channel. Meanwhile, the high CO_2_ affinity in IL increases the adsorption of CO_2_ of the filler, leading to a significant increase in selectivity. As shown in Figure 6a,b, for UiO-66-NH_2_@IL/PIM-1 MMMs under all loadings, the permeability of N_2_ and CH_4_ is less than that of pure PIM-1, indicating that gas transport of N_2_ and CH_4_ is indeed hindered in UiO-66-NH_2_@IL/PIM-1 MMM. As UiO-66-NH_2_@IL content increases, the permeability of N_2_ and CH_4_ of UiO-66-NH_2_@IL/PIM-1 MMM also increases slightly. The permeability of N_2_ increases from 367.6 Barrer to 626.5 Barrer and the permeability of CH_4_ increases from 671.9 Barrer to 1059.9 Barrer, respectively. Therefore, the selectivity is reduced correspondingly. CO_2_/N_2_ selectivity decreases from 22.5 to 16.0 and CO_2_/CH_4_ selectivity decreases from 12.3 to 9.5, respectively. This may be due to the excessive MOFs, resulting in a decrease in the bonding force between the filler/polymer phases. Similar phenomena can be observed in other literatures [3,10]. Nevertheless, the modified MMMs still have a higher selectivity than pure PIM-1 and far exceed the Robeson upper bound.

To further prove the effect of IL, MMMs with unmodified UiO-66-NH_2_ were conducted as a control experiment. For UiO-66-NH_2_/PIM-1 MMMs (Figure 6c,d), with the increase of UiO-66-NH_2_ filler, the permeability of all gases increased correspondingly. Specifically, at 30 wt.% loading, the N_2_ and CH_4_ permeability can reach 744.9 Barrer and 1425.4 Barrer, respectively, which is significantly higher than the permeability of UiO-66-NH_2_@IL/PIM-1 MMMs, so the selectivity of UiO-66-NH_2_/PIM-1 MMM has not been significantly improved. It is difficult to increase the permeability and selectivity of UiO-66-NH_2_/PIM-1 MMMs by simply adding MOF filler [3,10,35]. This may be the cause of interface defects due to the poor two-phase compatibility. Without the regulating function of IL, the larger size UiO-66-NH_2_ is difficult to evenly distribute in the PIM-1 matrix, so the packing particles will agglomerate and some non-selective bypasses will occur. Due to the existence of defects and bypass, the permeability of different gases can be increased, which ultimately leads to a sharp decrease in selectivity. Therefore, the IL coating strategy proposed in this work can effectively enhance the interfacial interaction between filler and polymer and reduce non-selective defects in MMMs. UiO-66-NH_2_@IL has better CO_2_ separation effect, which proves that the structure of UiO-66-NH_2_@IL/PIM-1 MMMs is more complete and there are fewer selective defects. The comprehensive test results are shown in Appendix A.

### 3.5. Aging Test

Due to the intrinsic characteristics of PIM-1 material, it is difficult to maintain high separation performance for a long time [41]. However, the modification of IL to the filler improved the interface compatibility and further made MMMs maintain a good separation effect after long-term use. 10 wt.% UiO-66-NH_2_@IL/PIM-1 MMM was selected as an example to test the permeability change of the membrane within 28 days (Figure 7). Throughout the testing process, the permeability of UiO-66-NH_2_@IL/PIM-1 MMM decreased slowly. The CO_2_ permeability changed from 8283.4 Barrer to 3264.0 Barrer in 21 days. After 21 days, the CO_2_ permeability still remained up to 3000–3300 Barrer. Accordingly, the CO_2_/N_2_ selectivity increased from 22.5 to 32.4, and the CO_2_/CH_4_ selectivity went up from 12.3 to 19.5 and remained basically constant after 21 days of aging. It was noticed that the performance of UiO-66-NH_2_@IL/PIM-1 MMMs steadily surpassed the Robeson upper bound, displaying excellent stability in gas separation [4,41].

### 3.6. Comparison of Membrane Properties

As shown in Figure 8, the prepared UiO-66-NH_2_@IL/PIM-1 MMMs broke through the Robeson upper bound very well both for CO_2_/N_2_ and CO_2_/CH_4_, far exceeding other polymer materials. Therefore, selecting PIM-1 as the gas membrane separation matrix is an effective strategy. Compared with other ternary system based on polymers of intrinsic microporosity and metal organic frameworks, our strategy is very different since the selected IL [bmim][Tf_2_N] can act as a suitable regulator of mixture in solvent immersion process to make uniform distribution of filler particles in MMMs, and amino-functional UiO-66 with abundant micropore structure can offer many additional pathway for CO_2_ permeation and form hydrogen bonds with PIM-1 to improve interfacial interactions, which has greatly reduced the non-selective defects of MMMs. Based on the adjustable compatibility between PIM-1 and MOF, the prepared mixed matrix membranes display better gas separation behavior for both CO_2_/N_2_ and CO_2_/CH_4_ compared with other PIM-based MMMs [9,16,21,34,42,43,44,45,46,47,48], displaying the wide applicability of stable and high-quality MMMs.

## 4. Conclusions

In summary, we report a new method to synthesize dense and defect-free amino-functional UiO-66/PIM-1 mixed matrix membranes by a coating modification and priming technique using the ionic liquid [bmim][Tf_2_N] as regulator. With this new method, MMMs with high filler content can be easily prepared. Both fluorine-containing ionic liquid and amino-functional UiO-66 had strong affinity towards CO_2_, enabling rapid and efficient penetration of CO_2_. Besides, with the introduction of oriented ionic liquids, the problem of agglomeration and leakage can be effectively solved, and the generation of voids and defects at the interface of MOF nanoparticles and PIM matrix can be eliminated. By decreasing the surface area and pore size of modified MOFs, N_2_ and CH_4_ can further be repelled to improve the molecular sieving effect. Different from traditional methods such as functional modification or graft modification, our strategy of IL coating is simple and efficient, and can be applied to different types of MOFs. In comparison with pure PIM-1 membrane, the gas separation performance of the prepared UiO-66-NH_2_@IL/PIM-1 MMMs can be greatly enhanced with larger CO_2_ permeance up to 8283.4 Barrer and competitively high CO_2_/N_2_ and CO_2_/CH_4_ separation factors of 22.5 and 12.3, respectively. In addition, our strategy demonstrates the applicability in synthesizing MMMs with excellent structural integrity and stability even under long-term operation, which opens a new venue for in-situ construction of high-quality MMMs.

## Figures and Tables

**Figure 1 membranes-11-00035-f001:**
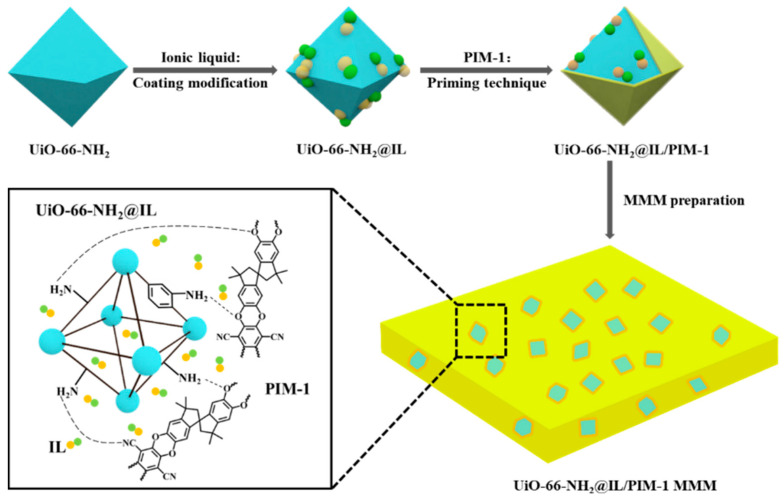
Schematic of the preparation process of UiO-66-NH_2_@IL/PIM-1 MMM by coating modification and priming technique.

**Figure 2 membranes-11-00035-f002:**
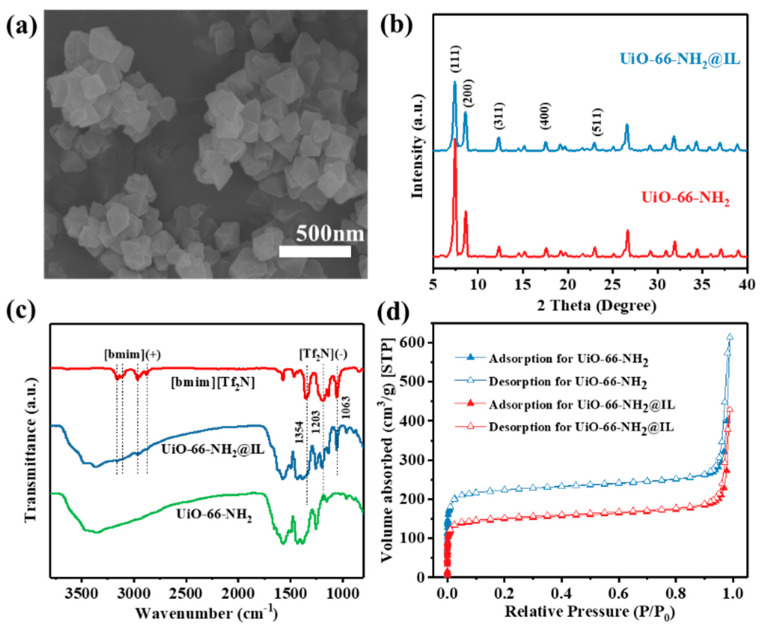
(**a**) SEM image of UiO-66-NH_2_@IL particles; (**b**) XRD patterns of UiO-66-NH_2_ and UiO-66-NH_2_@IL particles; (**c**) FT-IR spectra of [bmim][Tf_2_N], UiO-66-NH_2_ and UiO-66-NH_2_@IL; (**d**) N_2_ adsorption/desorption isotherm of UiO-66-NH_2_ and UiO-66-NH_2_@IL.

**Figure 3 membranes-11-00035-f003:**
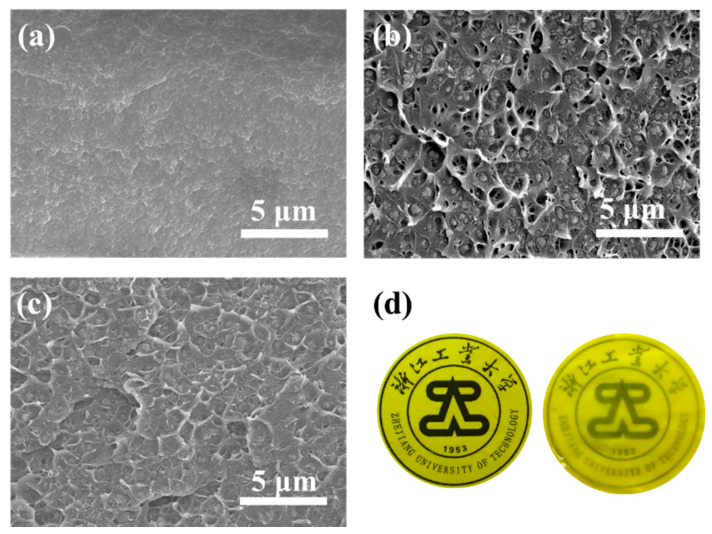
(**a**) SEM images of pure PIM-1 membrane; (**b**) UiO-66-NH_2_/PIM-1 MMM with 20 loadings; (**c**) UiO-66-NH_2_@IL/PIM-1 MMM with 20 loadings; (**d**) Photos of pure PIM-1 membrane (Left) and 20 wt.% UiO-66-NH_2_@IL MMM (Right).

**Figure 4 membranes-11-00035-f004:**
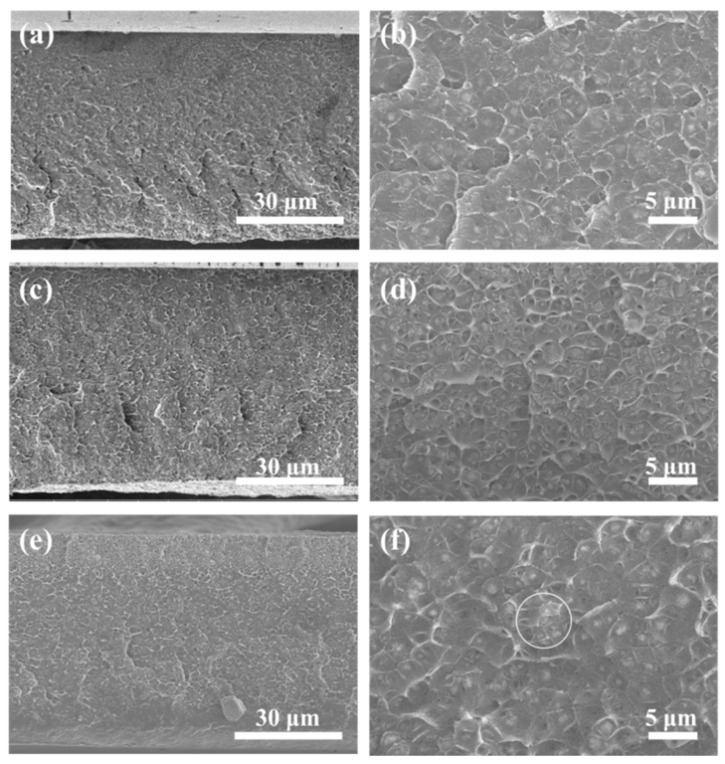
(**a**,**b**) Cross-section SEM images of UiO-66-NH_2_@IL/PIM-1 MMM with 10 wt.%, (**c**,**d**) 20 wt.%, and (**e**,**f**) 30 wt.% filler loadings.

**Figure 5 membranes-11-00035-f005:**
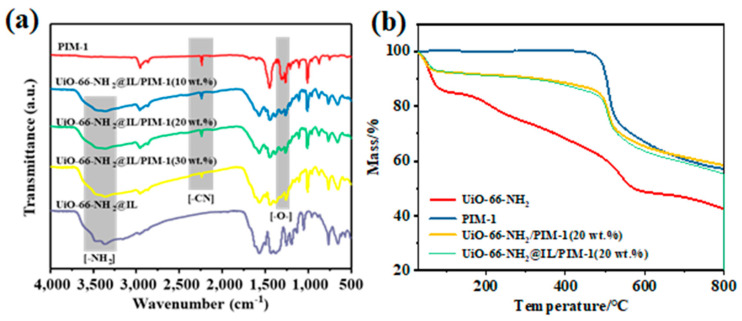
(**a**) FTIR spectra of pure PIM-1 membrane and UiO-66-NH_2_@IL/PIM-1 MMM, (**b**) TGA of UiO-66-NH_2_ powder, PIM-1 membrane, UiO-66-NH_2_/PIM-1 membrane and UiO-66-NH_2_@IL/PIM-1 membrane.

**Figure 6 membranes-11-00035-f006:**
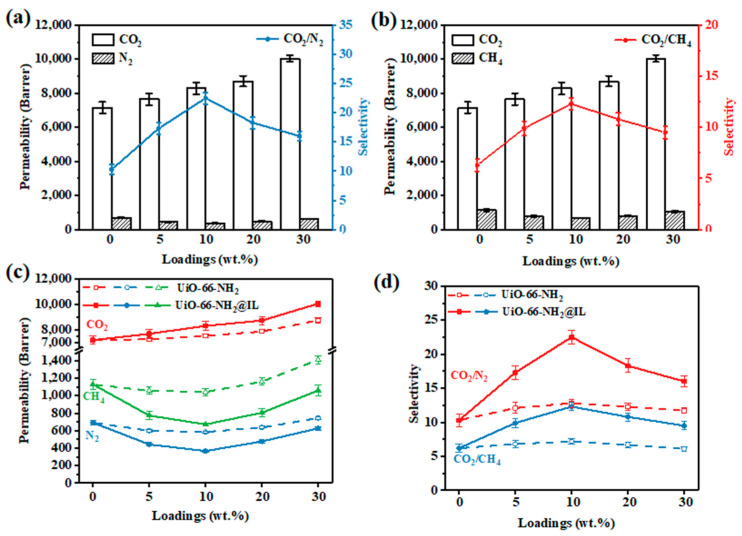
(**a**) CO_2_/N_2_ separation property of UiO-66-NH_2_@IL/PIM-1 MMM; (**b**) CO_2_/CH_4_ separation property of UiO-66-NH_2_@IL/PIM-1 MMM; (**c**) Pure gas permeabilities of MMM with different UiO-66-NH_2_ or UiO-66-NH_2_@IL loadings; (**d**) CO_2_/N_2_ and CO_2_/CH_4_ selectivities of MMMs with different UiO-66-NH_2_ or UiO-66-NH_2_@IL loadings.

**Figure 7 membranes-11-00035-f007:**
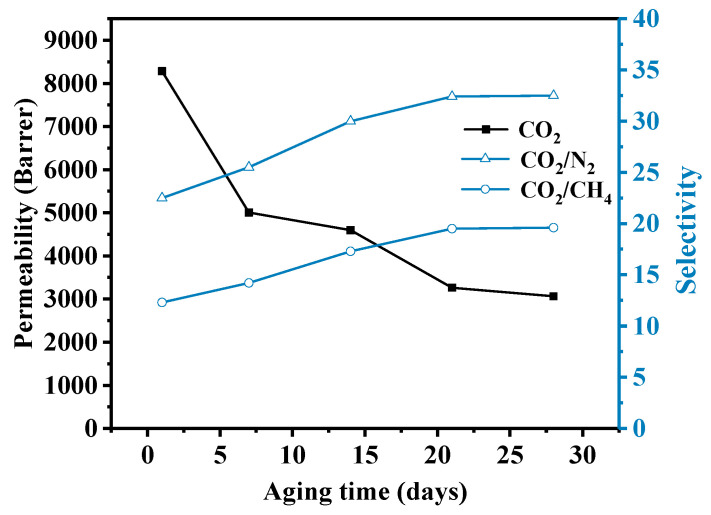
The separation performance of UiO-66-NH_2_@IL/PIM-1 MMM after aging tracked over 28 days.

**Figure 8 membranes-11-00035-f008:**
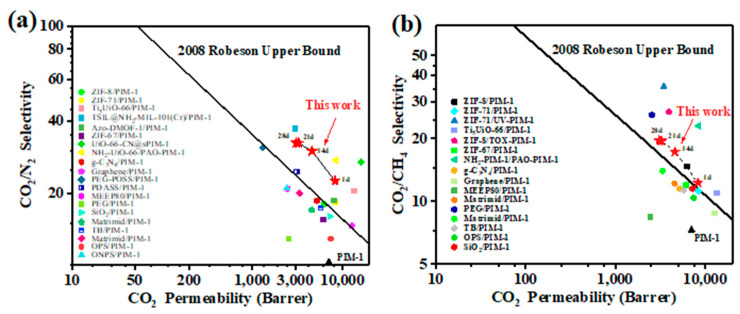
(**a**) Comparison of CO_2_/N_2_ performance between UiO-66-NH_2_@IL/PIM-1 MMM and other related MMMs; (**b**) Comparation of CO_2_/CH_4_ performance between UiO-66-NH_2_@IL/PIM-1 MMM and other related MMMs.

**Table 1 membranes-11-00035-t001:** Gas Sorption Data of UiO-66-NH_2_ and UiO-66-NH_2_@IL.

Sample	Surface Area (m^2^ g^−1^) [a]	Surface Area (m^2^ g^−1^) [b]	Pore Volume (mL g^−1^) [b]	Pore Width (nm) [b]
UiO-66-NH_2_	887.667	936.027	0.428	0.567
UiO-66-NH_2_@IL	585.413	610.922	0.307	0.524

[a] Analysis by MBET method; [b] Analysis by DFT method.

## Data Availability

Data is contained within the article or supplementary material.

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
