# Peer review of "Preparation of Amino-Functional UiO-66/PIMs Mixed Matrix Membranes with [bmim][Tf2N] as Regulator for Enhanced Gas Separation"

_membranes, 2021, doi:10.3390/membranes11010035_

Round 1

Reviewer 1 Report

This manuscript demonstrates the doping of modified MOFs fillers into PIMs membrane to enhance the gas separation properties. The results are well-presented and the paper is suitable for publication after the reviewer’s following concerns addressed:

  1. Given that the UiO-66-NH2 and IL were physically mixed to prepare UiO-66-NH2@IL, is it stable and if so, what interactions existed between two materials?
  2. How the authors expect the membrane performance with only IL doped in the PIMs?
  3. How about the performance when feeding gas mixtures?
  4. Since the gas selectivity was gradually decreased from 10 % to 30 wt% doping of fillers, have the authors tried a lower doping amount such as 5%, to see a higher selectivity?

Author Response

  1. Given that the UiO-66-NH2 and IL were physically mixed to prepare UiO-66-NH2@IL, is it stable and if so, what interactions existed between two materials?

Answer: Thanks for your comment. IL and UiO-66-NH2 were relatively stable. Most of [Bmim][Tf2N] were adsorbed and confined in the pores of UiO-66-NH2. The N and S atoms of [Tf2N]- formed a strong interaction with the metal node of MOF in the form of ionic bonds. This can also be confirmed from the report elsewhere. (Ban, YJ; Li, ZJ; Li, YS; Peng, Y.; Jin, H.; Jiao, WM; Guo, A.; Wang, P.; Yang, QY; Zhong, CL; Yang, WS Limitations of ionic liquids in nanocages: adjusting the molecular sieve properties of ZIF-8 for membrane-based CO2 capture. Angew. Chem. Int. Ed. 2015, 54, 15483-7.).

  1. How the authors expect the membrane performance with only IL doped in the PIMs?

Answer: Thank you for your comment. Due to the small amount of IL used in this work, we only tested the IL/PIM-1 membrane with a concentration of 10 wt%. After the preparation of the membrane, the IL/PIM-1 MMM was soaked in methanol before testing. The CO2 permeability of IL/PIM-1 membrane reached 11190.7 Barrer, the CO2/N2 selectivity was 13.4 and the CO2/CH4 selectivity was 7.2. IL/PIM-1 membrane had high permeability, but its selectivity was inferior to MOF/PIM-1 MMM.

  1. How about the performance when feeding gas mixtures?

Answer: Thank you for your comment. The pure gas test is also representative and can reflect the separation performance of the mixed matrix membrane. Many reports have selected this test method, for example, the research of Ma et al. (Ma, J.; Ying, Y.P.; Guo, X.Y; Huang, H.L.; Liu, D.H.; Zhong, C.L. Fabrication of mixed-matrix membrane containing metal–organic framework composite with task-specific ionic liquid for efficient CO2 separation. J. Mater. Chem. A 2016, 4, 7281-7288.) and Wang et al.(Wang, Z.G.; Wang, D.; Zhang, S.X.; Hu, L.; Jin, J. Interfacial design of mixed matrix membranes for improved gas separation performance. Adv. Mater. 2016, 28, 3399-3405.). Next in the future, we will employ practical mixed gas for test.

  1. 4. Since the gas selectivity was gradually decreased from 10 % to 30 wt% doping of fillers, have the authors tried a lower doping amount such as 5%, to see a higher selectivity?

Answer: Thank you for your comment. We have also tried mixed matrix membranes under a 5 wt% loading. As shown in the figure below, compared with pure PIM-1 membranes, the permeability of various gases has increased, but the selectivity is not as good as the membrane under 10 wt% loading.

Reviewer 2 Report

I recommend to add:

(1) The experimental conditions of GPC analyses together with a deeper discussion,

(2) TGA in air and discuss  the relationship  between the theoretical additive content and non-combustible residues (ash),

(3) The explanation in more detail how the additive with a given channel geometry can favourably affect the transport properties of gases with a given molecule sizes,

and to change the accuracy of the "BET area" values (xxx.xxx m2g-1). 

Author Response

Reviewer(s)' Comments to Author:

Reviewer #2:

  1. The experimental conditions of GPC analyses together with a deeper discussion.

Answer: Thank you for your suggestion. The experimental conditions of GPC analyses were added in the manuscript. In detail, polystyrene was used as the standard sample and tetrahydrofuran (1 mL min-1) was used as the mobile phase. The temperature of column and detector were 30 and 35 °C, respectively. We have included specific data about GPC in the supporting information. The molecular weight (Mw) of the material was 212.3 kDa and the number-average molecular weight (Mn) was 73.4 kDa, which were much higher than those reported in other literature (Zhao, S.; Liao, J.; Li, D.; Wang, X.; Li, N., Blending of compatible polymer of intrinsic microporosity (PIM-1) with Tröger's Base polymer for gas separation membranes. J. Membr. Sci. 2018, 566, 77-86.). The polydispersity Mw/Mn was 2.89.

Table S1. GPC test characterization results.

Distribution name

Mn/Da

Mw/Da

MP

Mz/Da

Mz+1/Da

PDI

Value

73405

212305

132713

720406

2001196

2.892240

  1. TGA in air and discuss the relationship between the theoretical additive content and non-combustible residues (ash).

Answer: Thank you for your comment. The TGA test was not performed in air. The TGA test was performed under a nitrogen atmosphere with the rate of 10°C/min. As can be seen from the Figure 5, thermal stability of PIMs matrix is much better than filler. Therefore, as the amount of filler increases, the amount of non-combustible residues will gradually decrease.

  1. The explanation in more detail how the additive with a given channel geometry can favourably affect the transport properties of gases with a given molecule sizes, and to change the accuracy of the "BET area" values (xxx.xxx m2 g-1). 

Answer: Thank you for your comment. It is generally believed that the additives with a given channel geometry will introduce new gas transmission channels for the originally dense PIMs membrane material. Then, the small-sized gas molecules can be easier to pass through the membrane, resulting in the increase of gas permeance. In addition, the pore size and volume decrease of UiO-66-NH2@IL may lead to increase the selectivity of UiO-66-NH2@IL/PIM-1 MMM.

Thank you for all your comments and suggestions!

12/22/2020

Reviewer 3 Report

The manuscript by Lu et al. presented a mixed matrix membrane based on PIM-1 and amino-UiO-66, pre-dispersed in ionic liquid to produce defect-free MMMs even at high loading. I am very pleased to see such work because personally, I think the method is one of the ‘current and highly potential’ approaches to solve the interfacial issues in the MMM study. I would recommend the paper to be ACCEPTED; however a few issues need to be addressed before the publication.

  1. Line 39-41. The statement mentioning ‘cryogenic liquefaction and solvent absorption have low efficiency’ is not correct. These technologies can produce the highest product purity (thus having the highest separation efficiency) compared to membrane technology. I suggest the authors correct that and mention other disadvantages instead.
  2. Line 44 – The latest upper bounds have been proposed for CO2/N2 and CO2/CH4 in 2019 (https://doi.org/10.1039/C9EE01384A ). I suggest to include/cite this too, and to use a more appropriate name for the upper bounds, such as ‘performance upper bounds or trade-off upper bounds.’
  3. Line 57 – 80. Please refer/cite this latest review on PIM-1 MMMs. The review is very focused on the use of nanomaterials to improve the performance and aging of PIM-1 (https://doi.org/10.1039/D0NR07042D ). I think it might help your discussion when talking about MMM approaches for PIM-1 and also when discussing physical aging.
  4. Line 87. Please delete the number ‘10’.
  5. Section 2.2: I do not see the step to remove the residual K2CO3 in your PIM-1 purification. So I would assume the produced PIM-1 will have impurities. The purity of PIM-1 can be determined/calculated from the NMR spectra. You might need to report this value.
  6. Section 2.3: Did you conduct the nanoparticles ‘activation’ before use? The activation is needed to ensure the framework pore is free from solvent and residual materials. Your high initial weight loss in TGA (Fig. 5b) is caused by this trapped solvent and NOT the nature of UiO-66-NH2 (as you mentioned in line 274: ‘UiO-66-NH2 have insufficient thermal stability’ – this statement is wrong). Please see https://doi.org/10.1016/j.memsci.2018.04.040. This paper discusses several types of UiO-66 and its thermal decomposition from TGA analysis in detail.
  7. Also, for info. Unactivated MOFs may show adequate adsorption capacity; however, this value is not the ‘true adsorption/potential’ value of the MOF. Some parts of pores are occupied with solvents and residues.
  8. Fig. 2(b). To include/compare the obtained MOFs with the parent UiO-66’s reference peak (https://doi.org/10.1021/cm1022882 ). Also, to discuss it accordingly in the text.
  9. Line 240-250: Your method of preparing MOF with IL is similar to the preparation procedure of Type III porous liquid (see this article in Nature, https://doi.org/10.1038/s41563-020-0764-y ). Even though you cannot present this as Type III porous liquid, the article may help you in further strengthen your discussion, such as (a) the IL provides stable MOF suspension solution and thus avoid pre-agglomeration, (b) avoid filler-polymer interfacial defects, and (3) increase the CO2 adsorption of the filler, contribute by the high CO2 affinity in the IL.
  10. The statements; Line 294-295 ‘These phenomena might be due to the fact that UiO-66-NH2@IL with narrow channels has a selective screening effect towards gas molecules’ and Line 298-299 ‘the introduction of IL further coated the outer surface of MOF and blocked the transport of other gases, leading to a significant increase in selectivity’ are not entirely correct. I suggest to improve it as Point #9.
  11. The CO2 permeability of UiO-66-NH2@IL/PIM-1 MMM changed from 8283 Barrer to 3264 Barrer. That is a 60% loss in just 21 days. I would say that it is very similar to the unmodified PIM-1 (please refer to https://doi.org/10.1039/D0NR07042D ). Do you have the aging values for the pristine PIM-1 as a comparison?
  12. The authors already presented several good and representative performance graphs and discussed them appropriately. So I think Table 2 is repetitive and to be moved to the supporting document.
  13. Fig. 8 – to include the current 2019 CO2/N2 and CO2/CH4 performance upper bounds (https://doi.org/10.1039/C9EE01384A ).
  14. Line 381 – Please update the supplementary materials information accordingly.
  15. To update the ‘Supplementary Information’ document. Authors’ names and affiliations are missing.
  16. General comment: one last read-through may need needed. Some sentences are unnecessarily complicated and may confuse the readers. Here are a few examples from the abstract:
    1. ‘… preparation of high-quality MMMs still keeps a big challenge due to lack of enough interfacial interaction between the two phases’….
    2. ‘… high-quality MMMs ignoring inevitably happened non-selective interfacial voids elsewhere…’

Author Response

Reviewer(s)' Comments to Author:

Reviewer #3:

  1. Line 39-41. The statement mentioningcryogenic liquefaction and solvent absorption have low efficiency is not correct. These technologies can produce the highest product purity (thus having the highest separation efficiency) compared to membrane technology. I suggest the authors correct that and mention other disadvantages instead.?

Answer: Thank you for your comment. We have revised this statement. “The currently used CO2 removal methods, such as cryogenic liquefaction and solvent absorption, usually require large amounts consumption of energy and cause environmental pollution.”

  1. Line 44-The latest upper bounds have been proposed for CO2/N2 and CO2/CH4 in 2019 (https://doi.org/10.1039/C9EE01384A). I suggest to include/cite this too, and to use a more appropriate name for the upper bounds, such as ‘performance upper bounds or trade-off upper bounds.

Answer: Thank you for your suggestion. This article detailed the potential of PIM membranes for industrial high-efficiency gas separation, and it had been cited it (Ref. 43). In spite of the fact that the latest upper bounds have been proposed for CO2/N2 and CO2/CH4 in 2019 (https://doi.org/10.1039/C9EE01384A), the 2008 Robeson upper bound is the most applied and widely recognized. The latest researches still adopt the 2008 Robeson upper bound, such as Yu et al. (Yu, G.; Zou, X.; Sun, L.; Liu, B.; Wang, Z.; Zhang, P.; Zhu, G., Constructing Connected Paths between UiO-66 and PIM-1 to Improve Membrane CO2 Separation with Crystal-Like Gas Selectivity. Adv. Mater. 2019, 31, 1806853.) and Qian et al.(Qian, Q.; Wu, A. X.; Chi, W. S.; Asinger, P. A.; Lin, S.; Hypsher, A.; Smith, Z. P., Mixed-Matrix Membranes Formed from Imide-Functionalized UiO-66-NH2 for Improved Interfacial Compatibility. ACS Appl. Mater. Interfaces 2019, 11, 31257-31269.)

  1. Line 57–80. Please refer/cite this latest review on PIM-1 MMMs. The review is very focused on the use of nanomaterials to improve the performance and aging of PIM-1 (https://doi.org/10.1039/D0NR07042D ). I think it might help your discussion when talking about MMM approaches for PIM-1 and also when discussing physical aging.

Answer: Thank you for your suggestion. The review focused on improving the physical aging of PIM-1. We have already cited the article (Ref. 40).

  1. 4. Line 87. Please delete the number ‘10’.

Answer: Thank you for your suggestion. We have corrected this error.

  1. Section 2.2: I do not see the step to remove the residual K2CO3 in your PIM-1 purification. So I would assume the produced PIM-1 will have impurities. The purity of PIM-1 can be determined/calculated from the NMR spectra. You might need to report this value.

Answer: Thank you for your comment. In fact, during the preparation of PIM-1, K2CO3 was not soluble in solvent. We only took out the yellow linear product from the three-necked flask, leaving a large amount of K2CO3 inside the container. After the preparation of PIM-1, deionized water (80 ℃) was used for multiple reflux washing, and the remaining K2CO3 will be removed later. And the data of NMR spectra was provided in the Supporting Information (Figure S1).

  1. Section 2.3: Did you conduct the nanoparticles ‘activation’ before use? The activation is needed to ensure the framework pore is free from solvent and residual materials. Your high initial weight loss in TGA (Fig. 5b) is caused by this trapped solvent and NOT the nature of UiO-66-NH2 (as you mentioned in line 274: nanoparti2 have insufficient thermal stability’ – this statement is wrong). Please see https://doi.org/10.1016/j.memsci.2018.04.040. This paper discusses several types of UiO-66 and its thermal decomposition from TGA analysis in detail.

Answer: Thank you for your comment. As mentioned in section 2.3, the MOF nanoparticles were activated before use. After MOF synthesis, DMF was used to wash 3 times and then methanol (60 ℃) was washed 3 times. Each washing process lasted 3 hours. After centrifugal filtration, the MOF particles were dried at 150 ℃ for 1 d. This process was to activate the as-prepared MOF. Compared with PIM-1, the thermal stability of UiO-66-NH2 particles is less insufficient. Our expression of this result in the manuscript was not clear enough, and the relevant sentences have been modified in the manuscript.

  1. Also, for info. Unactivated MOFs may show adequate adsorption capacity; however, this value is not the is not the alue is not the at 1502018.04.040. This paper discusses several types of UiO-66s and residues.

Answer: Thank you for your comment. The MOF nanoparticles were activated before use. After MOF synthesis, DMF was used to wash 3 times and then methanol (60 ℃) was washed 3 times. Each washing process lasted 3 hours. After centrifugal filtration, the MOF particles were dried at 150 ℃ for 1 d. This process was to activate the as-prepared MOF.

  1. 2(b). To include/compare the obtained MOFs with the parent UiO-66revious one for details.ere dried at 11021/cm1022882). Also, to discuss it accordingly in the text.

Answer: Thank you for your comment. We have already compared the obtained MOF with the parent UiO-66 in the given literature and found that there is no significant change in the position of the characteristic peak, and the article has been cited (Ref 20).

  1. Line 240-250: Your method of preparing MOF with IL is similar to the preparation procedure of Type III porous liquid (see this article in Nature, https://doi.org/10.1038/s41563-020-0764-y). Even though you cannot present this as Type III porous liquid, the article may help you in further strengthen your discussion, such as (a) the IL provides stable MOF suspension solution and thus avoid pre-agglomeration, (b) avoid filler-polymer interfacial defects, and (3) increase the CO2 adsorption of the filler, contribute by the high CO2 affinity in the IL.

Answer: Thank you for your suggestion. The solution proposed in this article can indeed strengthen our understanding of the role of IL. We have cited this paper and modified some of the content in the article.

  1. The statements; Line 294-295 is article can indeed strengthen our understanding of th2@IL with narrow channels has a selective screening effect towards gas molecules’ and Line 298-299 ‘the introduction of IL further coated the outer surface of MOF and blocked the transport of other gases, leading to a significant increase in selectivity’ are not entirely correct. I suggest to improve it as Point #9.

Answer: Thank you for your comment. Indeed, this content of the article contains some inappropriate errors. We have reviewed the manuscript and modified the corresponding sentence of this article.

  1. The CO2 permeability of UiO-66-NH2@IL/PIM-1 MMM changed from 8283 Barrer to 3264 Barrer. That is a 60% loss in just 21 days. I would say that it is very similar to the unmodified PIM-1 (please refer to https://doi.org/10.1039/D0NR07042D ). Do you have the aging values for the pristine PIM-1 as a comparison?

Answer: Thank you for your comment. In order to increase the permeability of the PIMs membrane, we have carried out methanol soaking treatment on the prepared membranes. However, the permeability of the methanol-treated membrane decreased rapidly and the aging phenomenon increased, which has been reported in the literature (Mitra, T.; Bhavsar, R.S.; Adams, D.J.; Budd, P.M.; Cooper, A.I. PIM-1 mixed matrix membranes for gas separations using cost-effective hypercrosslinked nanoparticle fillers. Chem. Commun. 2016, 52, 5581-5584.). In this work, the unmodified PIM-1 membrane was tested for aging, but the permeability dropped so fast that the value was too small, the data was not retained.

  1. The authors already presented several good and representative performance graphs and discussed them appropriately. So I think Table 2 is repetitive and to be moved to the supporting document.

Answer: Thank you for your suggestion. Table 2 has been moved to the Supporting Information.

  1. 8-to include the current 2019 CO2/N2 and CO2/CH4 performance upper bounds (https://doi.org/10.1039/C9EE01384A).

Answer: Thank you for your suggestion. This article detailed the potential of PIM membranes for industrial high-efficiency gas separation, and it had been cited it (Ref. 43). In spite of the fact that the latest upper bounds have been proposed for CO2/N2 and CO2/CH4 in 2019 (https://doi.org/10.1039/C9EE01384A), the 2008 Robeson upper bound is the most applied and widely recognized. The latest researches still adopt the 2008 Robeson upper bound, such as Yu et al. (Yu, G.; Zou, X.; Sun, L.; Liu, B.; Wang, Z.; Zhang, P.; Zhu, G., Constructing Connected Paths between UiO-66 and PIM-1 to Improve Membrane CO2 Separation with Crystal-Like Gas Selectivity. Adv. Mater. 2019, 31, 1806853.) and Qian et al.(Qian, Q.; Wu, A. X.; Chi, W. S.; Asinger, P. A.; Lin, S.; Hypsher, A.; Smith, Z. P., Mixed-Matrix Membranes Formed from Imide-Functionalized UiO-66-NH2 for Improved Interfacial Compatibility. ACS Appl. Mater. Interfaces 2019, 11, 31257-31269.)

  1. Line 381 Please update the supplementary materials information accordingly.

Answer: Thank you for your suggestion. We have updated the supplementary materials information.

  1. To update the tary materials informationls information acco names and affiliations are missing.

Answer: Thank you for your suggestion. We have supplemented relevant information.

  1. General comment: one last read-through may need needed. Some sentences are unnecessarily complicated and may confuse the readers. Here are a few examples from the abstract:

1.‘… preparation of high-quality MMMs still keeps a big challenge due to lack of enough interfacial interaction between the two phases’….

2.‘… high-quality MMMs ignoring inevitably happened non-selective interfacial voids elsewhere…’

Answer: Thank you for your suggestion. The manuscript has been reviewed and the confused sentences have been corrected.

Thank you for all your comments and suggestions!

12/22/2020

Round 2

Reviewer 1 Report

The authors have well-responded to my concerns.